# Analysis of Greek prehistoric combat in full body armour based on physiological principles: A series of studies using thematic analysis, human experiments, and numerical simulations

Andreas D. Flouris[1]*, Stavros B. Petmezas[1], Panagiotis I. Asimoglou[1], João P. Vale[2,3], Tiago S. Mayor[2,3], Giannis Giakas[4], Athanasios Z. Jamurtas[4], Yiannis Koutedakis[4,5], Ken Wardle[6], Diana Wardle[6]

1 Department of Exercise Science, FAME Laboratory, University of Thessaly, Volos, Greece, 2 Engineering Faculty, Transport Phenomena Research Centre (CEFT), Porto University, Rua Dr. Roberto Frias, Porto, Portugal, 3 Engineering Faculty, Associate Laboratory in Chemical Engineering (ALICE), Porto University, Rua Dr. Roberto Frias, Porto, Portugal, 4 School of Exercise Science, University of Thessaly, Volos, Greece, 5 Faculty of Education, Research Centre for Sport, Exercise and Performance, Health and Wellbeing, University of Wolverhampton, Wolverhampton, United Kingdom, 6 Department of Classics, Ancient History and Archaeology, University of Birmingham, Birmingham, United Kingdom

* andreasflouris@gmail.com, aflouris@uth.gr

**Data Availability Statement:** All data from this study has been made freely available and step-by-

## Abstract

One of the oldest complete suits of European armour was discovered in 1960 near the village of Dendra, in Southern Greece, but it remained unknown whether this armour was suitable for extended use in battle or was purely ceremonial. This had limited our understanding of the ancient Greek–Late Bronze Age–warfare and its consequences that have underpinned the social transformations of prehistoric Europe and Eastern Mediterranean. In a series of archeo-physiological studies, merging knowledge in archaeology, history, human physiology, and numerical simulation, we provide supporting evidence that the Mycenaean armour found at Dendra was entirely compatible with use in extended combat, and we provide a free software enabling simulation of Late Bronze Age warfare. A group of special armed-forces personnel wearing a replica of the Dendra armour were able to complete an 11-hour simulated Late Bronze Age combat protocol that we developed from a series of studies based on the available evidence. Numerical simulation of the thermal exchanges in Late Bronze Age warfare extended this conclusion across different environmental conditions and fighting intensities. Our results support the notion that the Mycenaeans had such a powerful impact in Eastern Mediterranean at least partly as a result of their armour technology.

## Introduction

Sixty years ago, a 3500 year old suit of bronze armour was discovered in a tomb near the village of Dendra, a few km from ancient Mycenae, in Southern Greece (Fig 1) [1]. It is considered as

step description of the rationale, methods, and results of this study is provided in an Online Supplement. A video showing the map of Homer's Troy and the surrounding area in 3D can be freely viewed online (https://youtu.be/jvQ9YTt6yzA) and has also been placed in an online data repository (https://doi.org/10.6084/m9.figshare.12961463.v1). The map was created using Azgaar's Fantasy Map Generator, a free web application, under a CC BY license, with permission from Max Haniyeu, original copyright 2017-2021. The data derived from the Iliad thematic analysis has been made freely available in an online data repository (https://doi.org/10.6084/m9.figshare.12961499.v1). Finally, an implementation of the Late Bronze Age Warrior model has been placed in an online data repository and is freely available for research and educational purposes (https://doi.org/10.6084/m9.figshare.12090831.v1).

**Funding:** The authors received no specific funding for this work.

**Competing interests:** The authors have declared that no competing interests exist.

one of the oldest complete suits of armour from the European Bronze Age (Online Supplement: Sections 1.1–3). Earlier experiments with replicas demonstrated its flexibility for use in combat but not its suitability for use in extended battle contexts [2–4]. Was it purely ceremonial [5,6]? This limited our understanding of ancient warfare, particularly in the Late Bronze Age, and its consequences which have underpinned the social transformations of the prehistoric world.

In the series of studies presented here, we have attempted to answer this research question and have provided supporting evidence to show that the armour discovered at Dendra was, indeed, entirely compatible with use in combat. A group of special armed-forces personnel wearing a replica of the armour were able to complete an 11-hour Late Bronze Age combat simulation protocol, developed on the basis of analyses of the Iliad. Numerical simulation of the thermal exchanges in Late Bronze Age warfare extended this conclusion across different environmental conditions and fighting intensities. To facilitate and promote research in this field, we developed a freely-available software enabling simulation of the thermal exchanges in Late Bronze Age warfare.

As no historical accounts or descriptions survive from the Greek Late Bronze Age regarding the scope and use of armour of the Dendra type, we turned to a key–and only–detailed early account of warfare, battle, and single combat: Homer's epic account of 10 days in the Trojan War, the Iliad. To suggest that Homer's epics were precise accounts of events which took place some five hundred years earlier, as Schliemann did in the 1870s, would invite both censure and disbelief. To follow Thucydides' (I.2-12) approach that the traditions about the remote past of the Greeks contain truth and suggest that they provide a reasonable basis for evaluating

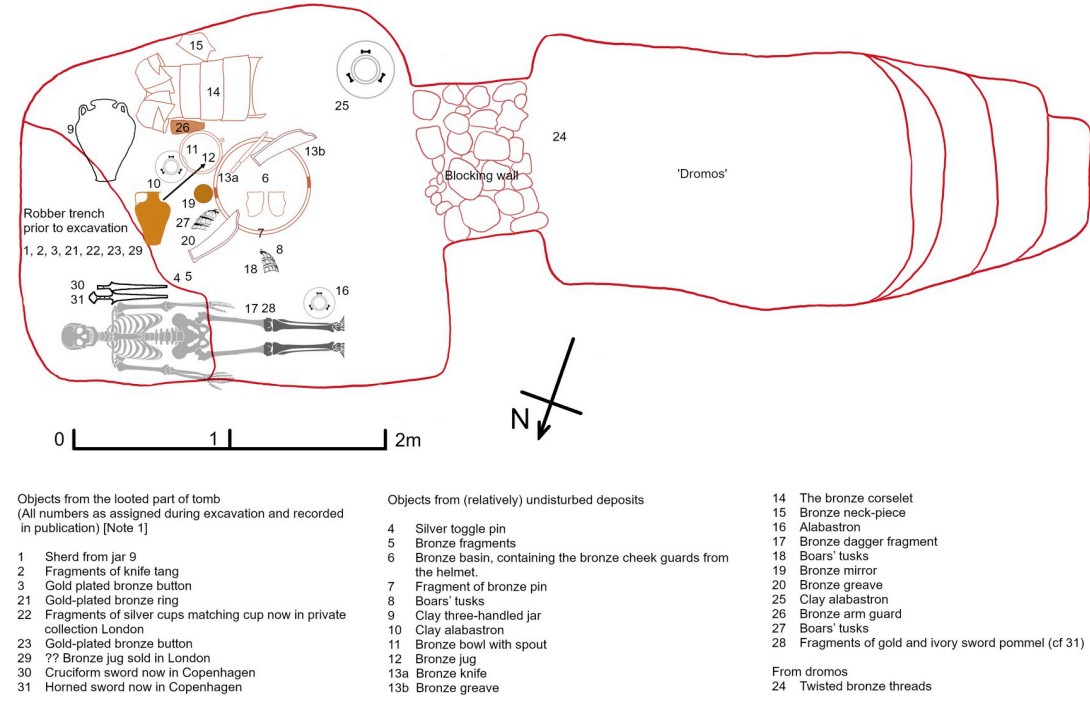

Objects from the looted part of tomb
(All numbers as assigned during excavation and recorded in publication) [Note 1]

| | |
|---|---|
| 1 | Sherd from jar 9 |
| 2 | Fragments of knife tang |
| 3 | Gold plated bronze button |
| 21 | Gold-plated bronze ring |
| 22 | Fragments of silver cups matching cup now in private collection London |
| 23 | Gold-plated bronze button |
| 29 | ?? Bronze jug sold in London |
| 30 | Cruciform sword now in Copenhagen |
| 31 | Horned sword now in Copenhagen |

Objects from (relatively) undisturbed deposits

| | |
|---|---|
| 4 | Silver toggle pin |
| 5 | Bronze fragments |
| 6 | Bronze basin, containing the bronze cheek guards from the helmet. |
| 7 | Fragment of bronze pin |
| 8 | Boars' tusks |
| 9 | Clay three-handled jar |
| 10 | Clay alabastron |
| 11 | Bronze bowl with spout |
| 12 | Bronze jug |
| 13a | Bronze knife |
| 13b | Bronze greave |

| | |
|---|---|
| 14 | The bronze corselet |
| 15 | Bronze neck-piece |
| 16 | Alabastron |
| 17 | Bronze dagger fragment |
| 18 | Boars' tusks |
| 19 | Bronze mirror |
| 20 | Bronze greave |
| 25 | Clay alabastron |
| 26 | Bronze arm guard |
| 27 | Boars' tusks |
| 28 | Fragments of gold and ivory sword pommel (cf 31) |

From dromos

| | |
|---|---|
| 24 | Twisted bronze threads |

**Fig 1. Diagram of the 'Cuirass' Tomb (no 12) in which the Dendra armour was found, illustrating the location of the offerings based on information [1] provided during the original excavation.** The chamber tomb is characteristic of Mycenaean civilization, consisting of an underground room with a long entrance corridor. The richly furnished tomb containing the warrior and his panoply was the smallest in a cemetery used for several centuries. An attempt by looters to rob the tomb early in 1960 was foiled, though not before some items had been removed.

the nature of early warfare in Greece is, we believe, a rational starting point. Accordingly, three independent reviewers conducted a literature review and two in-depth thematic analyses of Homer's Iliad and the extracted information was combined with physiological and biometeorological knowledge to create a combat simulation protocol replicating the daily activities performed by elite warriors in the Late Bronze Age. We hypothesised that the Dendra armour was compatible with use in combat if two conditions were fulfilled when this protocol was applied: (1) that the physiological strain caused by the armour was tolerable and within normal levels, and (2) that the armour did not limit the fighting ability of the wearer. A step-by-step description of the rationale, methods, and results of this study is provided in an Online Supplement where all our results and data are freely available.

## Materials and methods

### Studies 1–2: Analysis of Late Bronze Age warfare and battle tactics

The Trojan War, as described in the Iliad, was used as the archetype for Late Bronze Age warfare (Online Supplement: Sections 2.1–2.7). We extracted the information needed to create a Late Bronze Age combat simulation protocol by performing two thematic analyses complemented by a literature review. The 1st thematic analysis (termed Study 1) of the Iliad addressed the following five warfare and battle tactics topics: (1) the characteristics of the physical environment in which battles took place, (2) the typical start and end time of daily army operations, (3) the typical activities performed by warriors during one day, (4) the typical food and water intake during a day of battle, and (5) the physical characteristics and the level of combat experience of most warriors. The following additional topics were addressed in the 2nd thematic analysis of the Iliad (termed Study 2): (6) the main types of combat, and (7) the typical techniques, movements, and weapons used by combatants.

Studies 1 and 2 included a careful study of the Iliad performed independently by two reviewers who were either study investigators or had an extensive background in Classical studies. To eliminate inter-reviewer bias caused by interpretation of ancient Greek, the thematic analyses were performed using the Iliad translated into Modern Greek by Prof. Dimitrios N. Maronitis [7], a well-accepted translation that received the 2011 State Award for Interlanguage Translation. Notes addressing each topic were transcribed verbatim by a third investigator and were proofread by the original reviewer. Thematic analysis of transcripts was conducted independently by three investigators (ADF, SP, and PA). Thereafter, each investigator reviewed the others' results, and then they established the final themes by consensus.

Although Homer provides a detailed description of the weapons and combat moves in his epics [8], it is still unclear whether the Iliad provides a robust account of fighting tactics from the late Mycenaean period as it is a palimpsest of contents created in different periods. This results in, for example, problematic descriptions of armour and tactics in some cases. We navigated this problem and resolved contradictions by complementing our thematic analyses with an extensive review of the scholarly literature (Online Supplement: Sections 2.1–2.7). This ensured that the findings from the thematic analyses were in line with the best available evidence. For the literature review, we searched the Google Scholar and PubMed databases from their date of inception to January 2020 using search terms related to each of the above-mentioned seven warfare and battle tactics topics. We included all types of peer-reviewed papers, conference proceedings, and books. Manuscript titles and abstracts were subsequently screened for relevance. Thereafter, full texts (retrieved via the University of Thessaly library) were screened independently by three reviewers (ADF, SP, and PA). The results from the two thematic analyses of the Iliad combined with the results from the review of the literature are summarised in the Online Supplement (Section 2) and were used to create a Late Bronze Age

combat simulation protocol (Section 3), while the data has been placed in an online data repository and are freely available [9].

## Study 3: Human experiment to determine the physiological stress during Late Bronze Age warfare

We used the information from Studies 1–2 (Online Supplement: Sections 2.1–2.7) to conduct a human laboratory experiment where individuals underwent physiological measurements while performing the Late Bronze Age combat simulation protocol (S5 Table in S1 File). The study was performed in controlled environmental settings (Online Supplement: Section 4.1 and S19 Fig in S1 File) simulating the estimated ambient conditions (evaluated via a Kestrel 5400FW meter, Nielsen-Kellerman, Pennsylvania, USA) of the fighting described in the Iliad (Online Supplement: Section 2.1), while participants wore a replica of the Dendra armour and carried replicas of Late Bronze Age weapons. STROBE guidelines were applied in the reporting of the results from Study 3.

Thirteen males (Marines from the Hellenic Armed Forces; age: 29.2 ± 7.9 years; height: 1.73 ± 0.05 m; 74.1 ± 6.8 kg weight) volunteered to participate in Study 3. Data collection started in March 2019 and was completed within two months. They were selected to fit as much as possible the age and anthropometric characteristics of the elite warriors described in the Iliad (Online Supplement: Section 2.5). Prior to participating, they completed the Physical Activity Readiness Questionnaire to ensure that they presented with no health contraindications to performing extended strenuous physical exercise. They also read and signed an informed consent form approved by the institutional Ethics Review Board which approved the study, as well as an informed consent to publish the information/images/photographs taken during the study in an online open access publication. The experimental protocol adhered to the Declaration of Helsinki and was approved by the Bioethical Committee at the School of Exercise Science of the University of Thessaly (protocol number: 1098) as well as the Hellenic Army General Staff (Φ.300/74/449322 - Σ.2354—Athens, Oct. 19, 2017). The individuals appearing in Figs 2 and 7, S6, S16, and S17 Figs in S1 File in this manuscript have given written informed consent (as outlined in PLOS consent form) to publish these photographs. Extensive familiarisation for all methods was undertaken prior to the study (Online Supplement: Section 4.3) and all participants were free to cease their participation at any point before, during, or after the data collection and request that their data be removed.

Prior to undergoing any assessments, all participants were trained by the co-author SBP, a military officer and Licensed Instructor in Japanese fencing, martial arts, military combat tactics, and weapons (S6 Fig in S1 File). This 2-day training in groups focused on familiarisation and correct application of Late Bronze Age combat protection equipment and technology, correct and safe use of weaponry, as well as the different combinations of combat fighting techniques (close and middle range; S4 Table in S1 File) of the Late Bronze Age combat simulation protocol (S5 Table in S1 File). When necessary, additional individual training sessions were held. All participants underwent enough training to be deemed qualified by the Instructor for the unrestricted and safe use of the techniques, weapons, and armour used in the study.

Participants arrived at the laboratory for the main part of the investigation in the late afternoon (17:30–18:00) of the day prior to the day of testing. They were requested to abstain from caffeine since the morning of that day and they ate dinner (with *ad libitum* water) according to their weight-adjusted nutrition plan (Online Supplement: Section 2.4). Participants were requested to retire to bed at 22:00 and were woken at 05:30 according to the procedures outlined in Section 2 of the Online Supplement. At 06:00, they were provided with *ad libitum* water and a breakfast according to their weight-adjusted nutrition plan (Online Supplement:

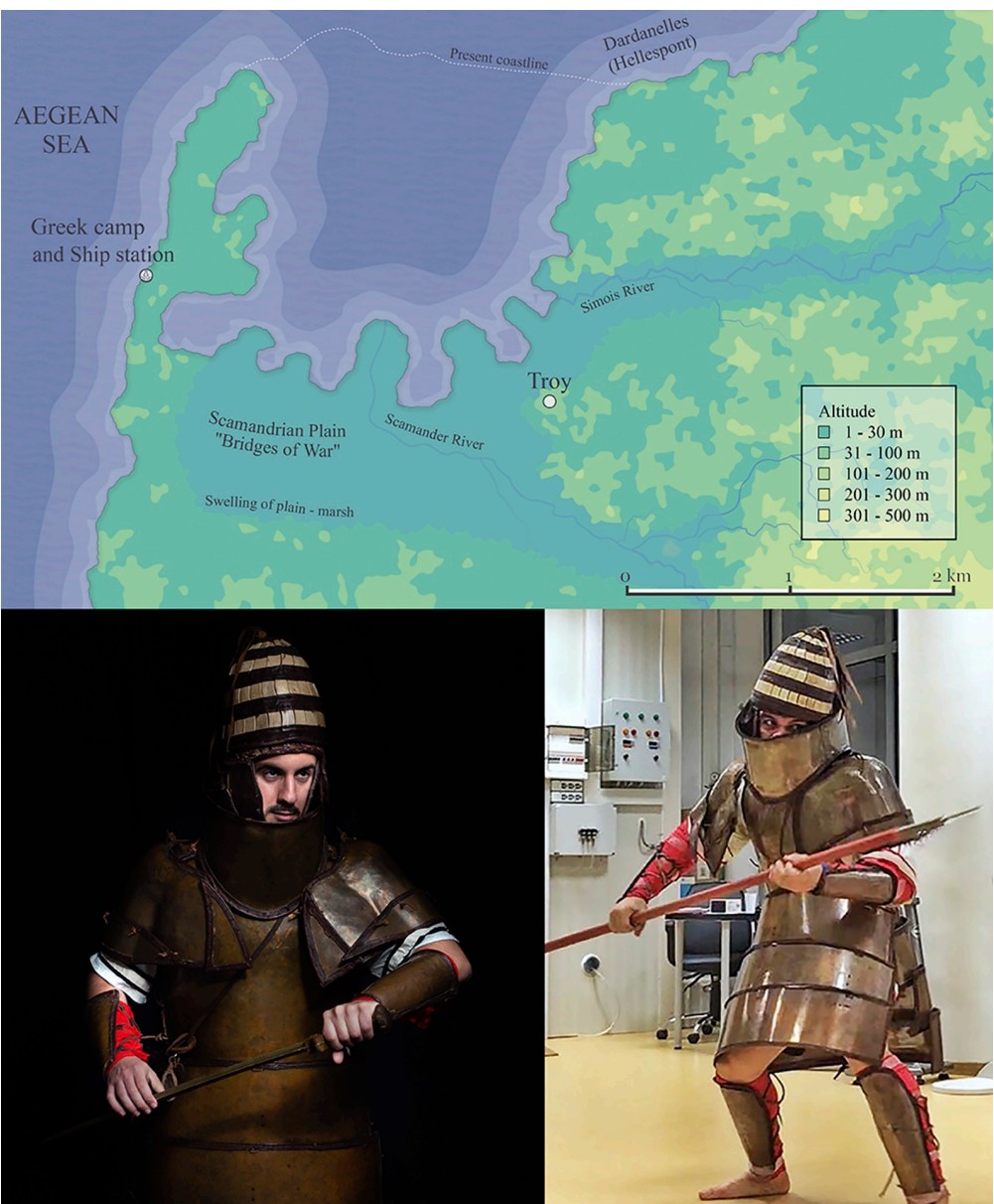

**Fig 2. Top: Geomorphology of the area surrounding Troy in the later phases of the Late Bronze Age (labels indicate the locations of the two army encampments and the geographic features of the area).** The map was created using Azgaar's Fantasy Map Generator, a free web application, under a CC BY license, with permission from Max Haniyeu, original copyright 2017–2021. Bottom: Volunteer marine soldiers in simulated combat wearing the Dendra armour replica during the empirical study (right) and an artistic photo shoot (left). Photo credit: Andreas Flouris and Marija Marković. Permission required for reproduction.

Section 2.4). At 06:30, volunteers were weighed and gave a urine sample to determine hydration status. Thereafter, all sensors were placed, and a blood sample was taken at 06:55. The Late Bronze Age combat simulation protocol (S5 Table in S1 File) was initiated precisely at 07:00 and timing was followed very carefully throughout until its completion, at 17:54.

During the various phases of the Late Bronze Age combat simulation protocol, participants performed the combat move combinations outlined in S4 Table in S1 File and were continually encouraged to exert maximal effort. During the breaks included in the combat simulation

protocol, participants were allowed to consume *ad libitum* water as well as snacks according to their weight-adjusted nutrition plan (Online Supplement: Section 2.4). Once the protocol was completed, participants gave another set of urine and blood samples. Also, the following parameters were assessed before, at specific times during (S5 Table in S1 File), and immediately after the combat simulation protocol: blood glucose concentration, blood lactate levels, hand grip strength, reaction time to visual and auditory stimuli, as well as self-perceived rate of exhaustion, thermal comfort, and thermal sensation. Heart rate, core body temperature, and mean skin temperature were continuously assessed throughout the combat protocol. The energy cost of each activity in the combat protocol was assessed via a portable gas analyser. For the latter measurement, we randomly evaluated specific parts of the combat simulation protocol for some participants to avoid interrupting the flow of combat and thus limiting the participants' maximum effort.

For the purposes of this study, we used a replica of the Dendra armour (S9 Fig in S1 File) made in Birmingham, UK. [10], in 1984 by staff and students of the Metal-working Department at the Bournville College of Art in Birmingham under the direction of Diana Wardle (Online Supplement: Section 4.5). The replica was made using gilding metal (95% copper, 5% zinc), which was the closest alloy to the original bronze available. Each of the plates of the armour followed exactly the dimensions, curvature, and perforations of the original and was edged with leather strips and joined with leather thongs. The helmet plaques imitating the original boar's tusk helmet were made from cast resin by H. Buglass of the former Department of Ancient History and Archaeology, University of Birmingham, and the leather inner cap, as described by Homer [11], was plaited by Ken Wardle. The total (armour and helmet) weight of the replica was 23.32 kg [measured with a precision weight scale (Kern DE 150K2D, Kern & Sohn GmbH, Balingen, Germany)], as opposed to the original which is estimated to weigh about 18 kg (missing one handguard, parts of greaves, and after 3500 years of oxidisation). The weight of each piece of the replica is provided in S9 Fig in S1 File. Recent tests of the efficacy of such armour have revealed that it offered considerable protection against most blows [5]. Throughout the manuscript, we refer to the replica as the Dendra armour.

For the purposes of this study, we created a replica Mycenaean cruciform sword (Sandars type Di) [12] of the type reported to have been found in the Dendra tomb, which had blunt edges and point for safety reasons using methods and materials of the archaic period (Online Supplement: Section 4.6 and S10 Fig in S1 File). The sword was 0.85 m long with a weight of 1.2 kg (copper blade, wooden handle, copper ornamental nails and pins), being slightly heavier than the original. For the combat moves requiring spear throw or strike, we used a 2.2 m replica wooden (ashwood) spear with a copper head blunt edges and point for safety reasons (0.46 kg), similar to those described in the Iliad. Also, a medium-sized stone (23 x 15 cm; 1.3 kg weight) was used in the "foot warrior vs chariot" encounters (S4 Table in S1 File) and a composite recurve bow [13] was used to shoot an arrow at close-range in the "chariot vs warrior on ship" encounter (S4 Table in S1 File).

## Study 4: Numerical study using a Late Bronze Age Warrior model

To investigate whether the Dendra armour was compatible with combat use for different environmental conditions (including e.g., solar exposure) and fighting intensities (i.e., metabolic rate), we developed a numerical model, henceforth called the Late Bronze Age Warrior model, simulating the thermoregulatory system of the elite Late Bronze Age warrior wearing Dendra-type armour (Online Supplement: Section 5). The Late Bronze Age Warrior model was developed starting from the Tanabe 65-node model [14], to which we incorporated several modifications to account for phenomena that are key for the goal of this study. To provide a detailed

account of the sun-induced radiant exchanges of a virtual subject performing physical activity while wearing Dendra-like armour (instead of typical clothing considered in the Tanabe 65-node model), we incorporated modifications to the model code describing more accurately the effects of the clothing/armour and the radiant exchanges on the body's thermal exchanges with the environment. These modifications were implemented to develop the Late Bronze Age Warrior model and its predictions were compared, together with those obtained with the Tanabe 65-node model, with results from several independent experimental studies (Online Supplement: Section 5 and S16 Fig in S1 File). An implementation of the Late Bronze Age Warrior model has been placed in an online data repository [15] and is freely available for research and educational purposes.

## Results and discussion

### Location, date, and environment

Our thematic analyses combined with published sedimentology and geomorphology data [16] suggest that the Trojan War took place in a ~4 km² area dominated by the Scamander river plain, with meandering river levees and backswamps, and a coastline with irregular shaped bays and pervasive coastal marshes. The battleground morphology was characterised by flat ground with low hills of maximum 30 m altitude. By combining this data [16], the locations of morphology and historic features of the Iliad deduced by Luce [17], and the geographic indicators and distances provided by Strabo [18], we used Azgaar's Fantasy Map Generator, a free web application (under a CC BY license, with permission from Max Haniyeu, original copyright 2017–2021), to create our interpretation of the geomorphologies in the area surrounding Troy at the time of the events described in the Iliad (Fig 2). A video showing the map of Homer's Troy and the surrounding area in 3D is available at https://youtu.be/jvQ9YTt6yzA.

Dating the Trojan War was outside the scope of the present study, yet setting a hypothesized time period and season was needed to calibrate the environmental conditions for our experiments (temperature, day-night cycle, etc.). Our thematic analyses of the Iliad suggested that the events described took place in the summertime. Assigning a date for the Trojan War is a complex challenge (see Limitations section) which has been a subject of frequent discussion since antiquity. Ancient authors have placed the Trojan War within a long period between 1334 and 1129 BC [19–23], yet it is not clear whether later narratives are reliable evidence for dating the event [24,25]. Considering this information, we hypothesised that the events described in the Iliad took place in June during the later phases of the Late Bronze Age.

Paleoclimate data using radiocarbon dating of stable oxygen and carbon isotopes in samples from the surrounding region [26] coupled with palaeoceanographic research using the regional marine sediment core [27–29] suggests that the average annual temperature in the Troad during the later phases of the Late Bronze Age was 18–20°C and the average annual relative humidity was 70–80%. Assuming a monthly variability similar to that of the present (1980–2017) [30], we estimated a temperature of 24–29°C and a relative humidity of 70–85% in June.

### Daily activities and nutrition of the warriors

Our thematic analyses indicated that armies moved out of encampment ~2.5 hours after sunrise and that fighting ended ~1.5 hours before sunset. The average sunrise and sunset times in the Troad during June in the later phases of the Late Bronze Age were 04:37 and 19:37, respectively (±10 minutes) [31]. Based on these figures, we estimated that army operations started at ~07:00 and ended at ~18:00, which confirms previous analyses of Homeric [32] and Classical Greek [33,34] warfare. We used these assumptions–together with geographic indicators and

distances provided by Strabo [18], lithosome distributions and radiocarbon dating [16], estimates of marching speed of armies in Classical Greece [34,35], and day-by-day thematic analysis of the battles described in the Iliad (S3 Fig in S1 File)–to estimate the duration (S1 Table in S1 File), timing (S5 Table in S1 File), and energy cost (S2 Table in S1 File) of the activities performed by elite warriors, such as the one who had worn the Dendra armour.

Assuming that Late Bronze Age warriors consumed enough food to account for the energy requirements of the daily battle activities (estimated as 4,443 kcal; Online Supplement: Section 2.4), we developed a nutrition plan (S3 Table in S1 File) providing our soldiers with appropriate amounts of energy for the strenuous combat activities included in the Late Bronze Age combat simulation protocol. Our thematic analyses suggested that ~40% of Homeric warriors' daily caloric intake was provided by the morning breakfast, ~10% was consumed during fighting breaks throughout the day, and the remaining ~50% was provided by the evening dinner– a distribution adopted in our study's nutrition plan (Online Supplement: Section 2.4). The food selection and preparation in our study were chosen to match the food available/consumed in the Iliad [36–39], the Homeric dietary and culinary practices [36–39], published biomolecular archaeology data of stable isotope analyses of human and faunal remains in the region [40], as well as archaeological data on food production and the consumption of stock-farming animal species [41–43]. Accordingly, breakfast was comprised primarily of dry bread, goat's cheese, olives, and red wine. The small snacks consumed *ad libitum* during the day included dry bread, honey, goat's cheese, and onions. Finally, meat (sheep/goat/pig/cow) and red wine were the main components of the dinner at the day's end, but bread and cheese were also included.

## Warrior characteristics

Homer describes the majority of warriors as strong, fast, and skilled mature men. Also, many of the elite warriors are described as tall and physically impressive (Online Supplement: Section 2.5). While it is logical to suspect that the physique of epic warriors was exaggerated and embellished in Homer's oral poetry, archaeological evidence [44–46] indicates that warriors were likely to be larger and stronger than the average population. Skeletal remains from the Late Bronze Age suggest that the mean height of the average Greek male was 1.62–1.67 m [45,47], however that of the elite males buried in Grave Circle B at Mycenae averaged 1.7 m [43], while the height of the wearer of the Dendra armour was >1.77 m [1]. Therefore, we selected for our study Marines from the Hellenic Armed Forces fitting these age (29.2±7.9 years) and anthropometric characteristics (1.73±0.05 m height; 74.1±6.8 kg mass).

## Combat units, weaponry and tactics

We analysed the Iliad to design a combat protocol simulating the daily activities of elite warriors in the Late Bronze Age. Our thematic analyses revealed that the army was organised in different cooperating groups composed of leaders surrounded by their followers (Online Supplement: Sections 2.3 and 2.6). Leaders wore full, well-made, and functional armour and were typically elite warriors with extensive battle experience. The majority of followers wore light or no armour and were capable warriors with varying levels of skill, training, and fighting experience. The groups constantly attacked, retreated, recovered, gained ground, and retreated again. At any given time during battle, a number of elite warriors with their followers fought at the front line as "*foremost fighters*", while others (referred to as "*multitude*") would seek safety at the rear of the battle line (Online Supplement: Sections 2.3 and 2.6). In total, the Homeric fighting activity was characterised by hit-and-run tactics [25,32,48], a form of physical effort described in physiology as "high-intensity interval exercise" [49].

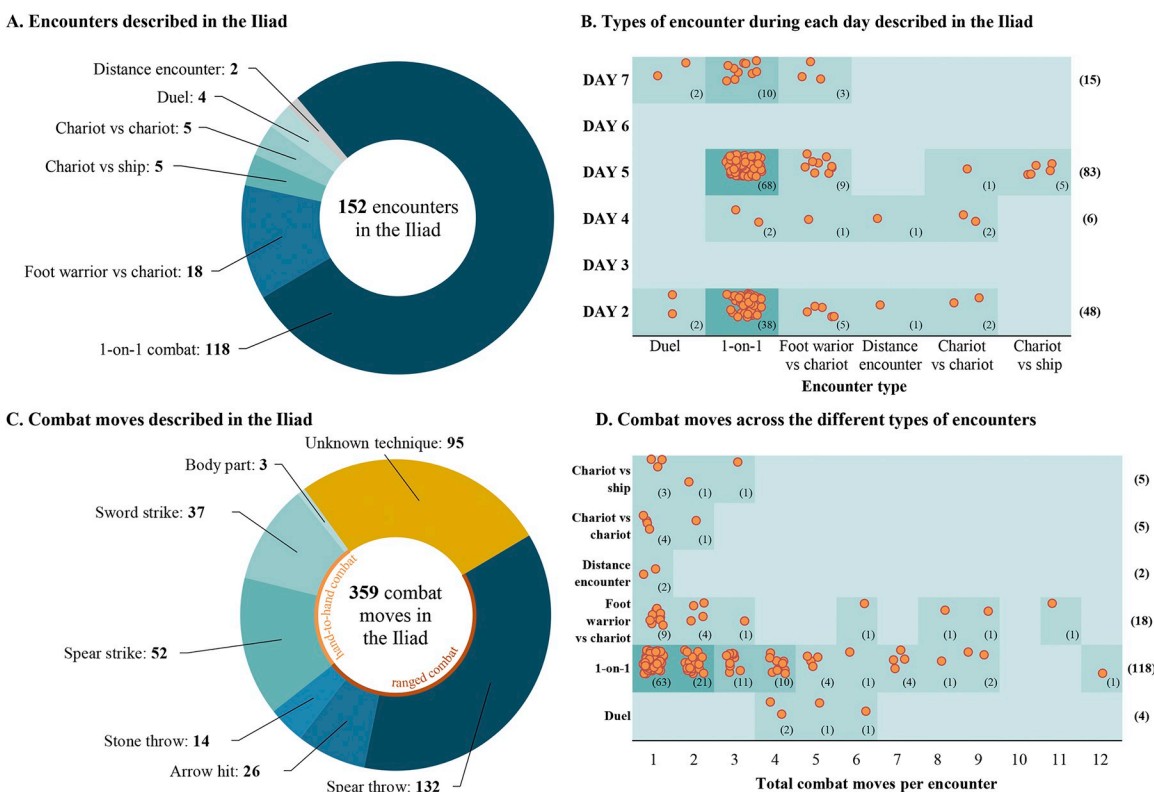

**Fig 3.** The total (panel A) and daily (panel B) encounters described in the Iliad as well as the combat moves used by warriors (panel C) across the different days described in the Iliad (panel D). Panels B and D are two-level scatterplots with data points randomly distributed within each rectangle at the crossing of the horizontal and vertical axes. The colour of rectangles in the scatterplots indicates the frequency of data points included (darker shades indicate more data points).

The identified hit-and-run tactics and high-intensity-interval exercises as well as a day-by-day thematic analysis of the battles described in the Iliad (Online Supplement: Section 2) were taken into consideration to design a Late Bronze Age combat simulation protocol describing the activities undertaken by elite warriors during daily army operations which, as explained above, started at ~07:00 and ended at ~18:00. To develop this 11-hour combat simulation protocol, we used a two-step cluster analysis of the combat moves described by Homer to identify the typical number of moves, weapons used, phases (attack-defence), and areas of the body targeted (S4 Table in S1 File). The Iliad includes 152 encounters (Fig 3A), 78% of which being "one-on-one" combats (i.e., foot warrior vs foot warrior; Fig 3B). A total of 264 combat moves are described in detail (Fig 3C), of which 65% involved ranged weapons (where spears or even stones were thrown) and 35% involved fighting at close quarters combining different moves and weapons (Fig 3D). Eighty percent of the encounters ended within three combat moves and 53% ended in a single fatal move (S4 Fig in S1 File). The chest, the head, and the upper limbs were the most commonly targeted areas of the body (S5 Fig in S1 File). The developed Late Bronze Age combat simulation protocol appears in S5 Table in S1 File.

## Human experiment

The study followed precisely the times, nutrition, movements, and combat tactics described above.

All participants successfully completed the 11-hour Late Bronze Age combat simulation protocol, with the majority developing reactive leucocytosis–at levels typically observed

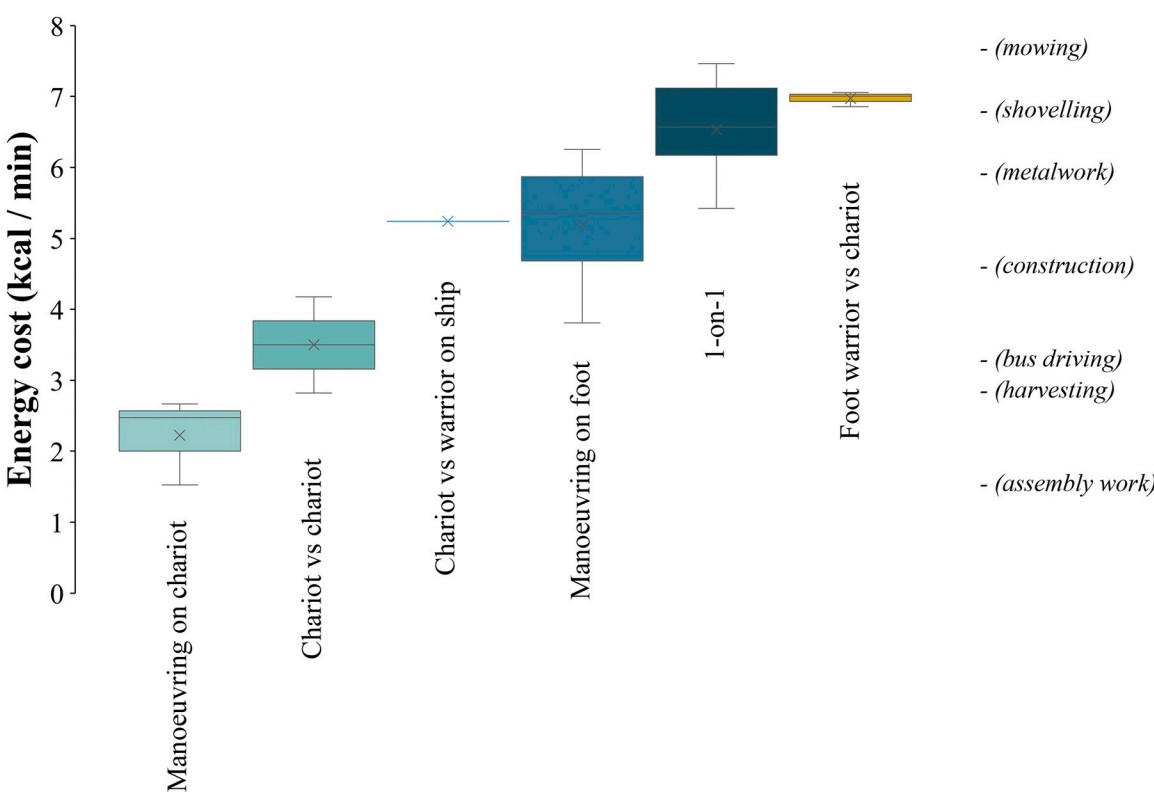

**Fig 4. Box plot showing the energy cost from indirect calorimetry during different activities of the Late Bronze Age combat simulation protocol.** For each activity, the top and bottom whiskers represent the maximum and minimum values, the top and bottom of the coloured rectangle represent the 3rd and 1st quartile, the lines represent the median, and the X marks represent the mean. The energy cost of various activities [51] is listed on the right side of the graph as reference. The effect size of the differences for all comparisons was large (d ≥ 0.8) except for between "1-on-1" and "Foot warrior vs chariot".

following extreme stress and exercise–due to the level of physiological strain endured (S6 Table in S1 File) [50]. They also showed a high level of fatigue, sore upper body due to the weight of the armour, and foot pain due to walking, running, riding a chariot, and fighting barefoot. The "one-on-one" and "foot warrior vs chariot" encounters were the most intense activities, while "manoeuvring on chariot" and the "chariot vs chariot" encounters were the least intense (Fig 4). Core body (36.4–37.7˚C) and mean skin (29.1–36.1˚C) temperatures indicated minor hyperthermia, while heart rate (57–166 beats/min) suggested a moderate-to-high effort during simulated combat (up to ~80% of maximum heart rate; Fig 5 and S11 Fig in S1 File).

The force generated by participants during combat hits was relatively high (3.5±0.9 kN; Fig 6 and S12 Fig in S1 File) and could inflict serious injury. Previous studies have shown that fracture risk begins at any stroke with >2 kN impact force [52] (i.e., >99% of hits in our study), while hits between 3.3 [53] and 4.8 [54] kN (i.e., >89% of hits in our study) lead to severe fractures. There was a moderate decline in blood glucose levels across time (r = -0.234, p = 0.017), from an average of 105.5±14.0 mg/dL in the morning to 89.4±9.0 mg/dL at the end of the 11-hour combat simulation protocol (S13 Fig in S1 File). The Iliad-based nutrition plan provided enough energy to complete the combat simulation protocol, since hypoglycemia (< 70 mg/dL blood glucose) at the day's end was only observed in one of the 13 volunteers. Measurements of hand grip strength, reaction time to visual and aural stimuli, hydration levels (via urine specific gravity), blood lactate, interleukin 6 and 10, as well as tumour necrosis factor

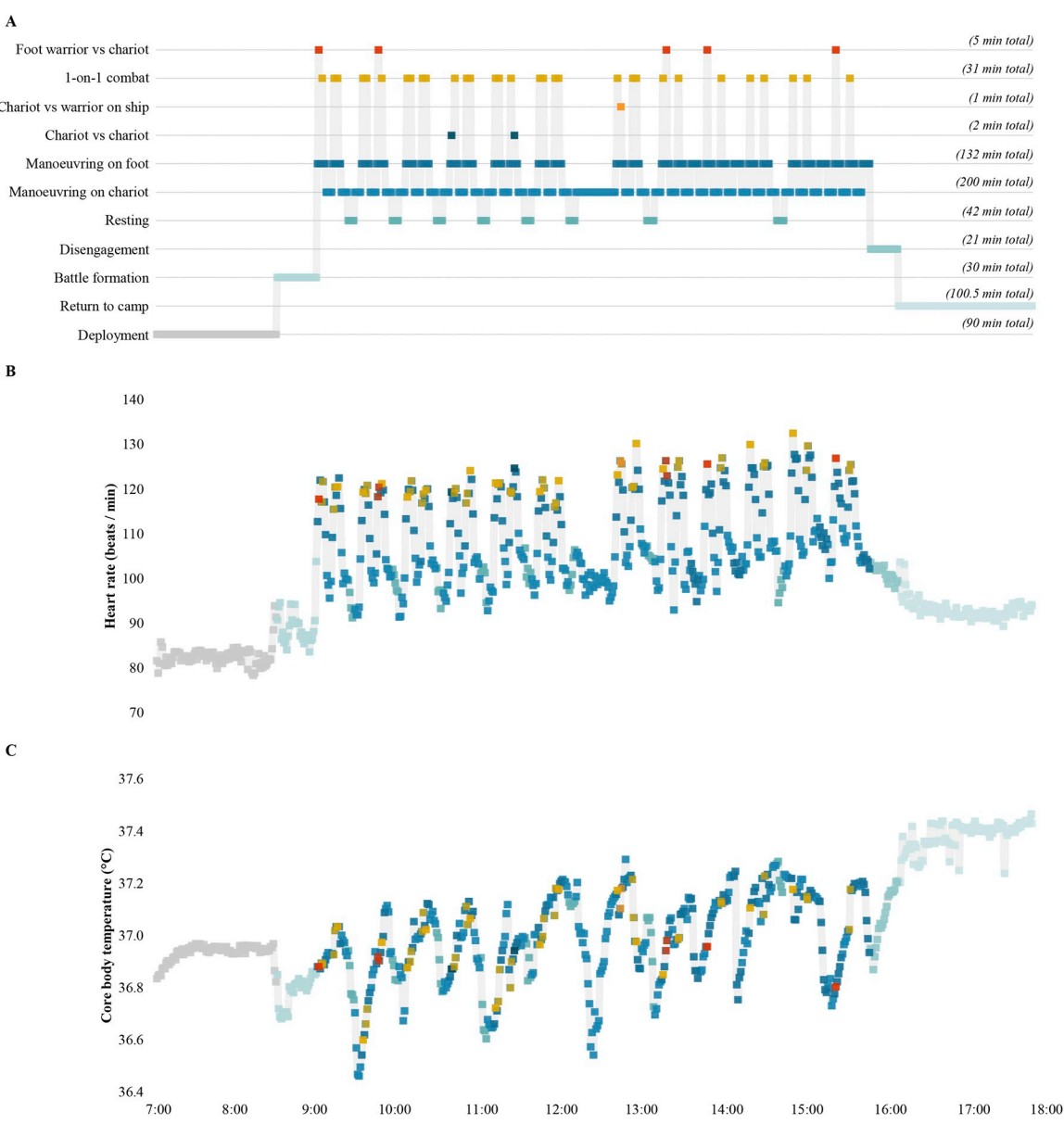

**Fig 5.** The Late Bronze Age combat simulation protocol incorporating the activities performed by elite warriors during one day described in the Iliad (panel A) and the average heart rate (panel B) and core body temperature (panel C) of the volunteers undergoing this physical task. Colours indicate the different stages of the combat protocol.

alpha showed no significant changes throughout the protocol (p>0.05; Online Supplement: Section 4.8, S13 Fig and S6 Table in S1 File). Taken together, these experimental results indicate that the Dendra armour was compatible with use in the type of combat described by Homer in the Iliad.

Following the end of data collection, a photo shoot was organised for the purpose of creating a documentary visual account of the measurements (Fig 7 and S14 and S15 Figs in S1 File). In this case, the sensors and measuring equipment were removed as the primary focus was on the warrior, the armour, the weapons, the body postures and the combat moves.

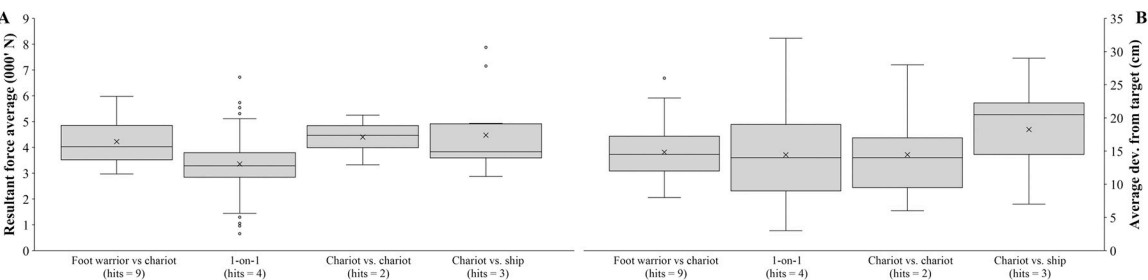

**Fig 6.** Characteristics of the resultant force generated (graph A) and the deviation from the target centre (graph B) during hits across the different encounter types. Shaded areas represent standard deviation. Note: a = different from foot warrior vs chariot ($p < 0.05$); b = different from 1-on-1 ($p < 0.05$); c = different from chariot vs chariot ($p < 0.05$); d = different from chariot vs warrior on ship ($p < 0.05$).

## Numerical simulation study

Our modelling work (Online Supplement: Section 5) started from the Tanabe 65-node model that predicts the body temperature of virtual individuals and their heat/mass exchanges with the surrounding environment [14]. We then implemented several modifications to strengthen the model accuracy regarding phenomena that were key for this study, including (1) the effects of the armour on the body thermal exchanges with the environment, (2) the calculation of the long-wave radiation exchanges, (3) the effect of solar loads varying with the sun position in the sky, and (4) the energy lost by work during intense activities (important for high metabolic rate activities). The developed Late Bronze Age Warrior model has been incorporated into a freely available software placed in an online data repository and its validity was confirmed as the predicted core body temperatures were found to be within 0.1–0.2°C of that in the above-mentioned human experiment (S23 Fig in S1 File).

We then considered seven simulation cases (S11 Table in S1 File) to investigate how different combinations of air temperature, wind, solar exposure, fighting intensity, and use of a white surcoat over the armour impacted the core body temperature of the warrior as well as his heat/mass exchanges with the surrounding environment. Our underlying premise was that the warrior would stop fighting if his core body temperature surpassed 38.5°C–a conservative threshold as modern athletes often surpass that level without harm [55]. We found that all but one of the tested conditions would allow the warrior to continue fighting for the entire 11-hour combat simulation protocol (S24 Fig in S1 File). In the remaining test condition–an improbable case with unrealistically low wind speed, high temperature, and a very high fighting intensity–the warrior would be forced to stop fighting after 7.5 hours into battle. As a whole, the numerical analysis indicated that the Dendra armour was compatible with combat use for different environmental conditions and fighting intensities.

## Limitations

Despite our best intentions and extensive efforts, our four studies have limitations that should be considered when evaluating their results. Due to these limitations, our findings must be supplemented with historical analysis and theoretical modelling to gain a more comprehensive understanding of Late Bronze Age warfare.

First, the Iliad is undoubtedly the most appropriate text to use for developing a Late Bronze Age combat simulation protocol [8,25]. However, it is not clear whether it provides a robust account of fighting tactics from the late Mycenaean period, as it is a palimpsest of contents created in different periods and can lead, for example, to misinterpretation or oversimplification

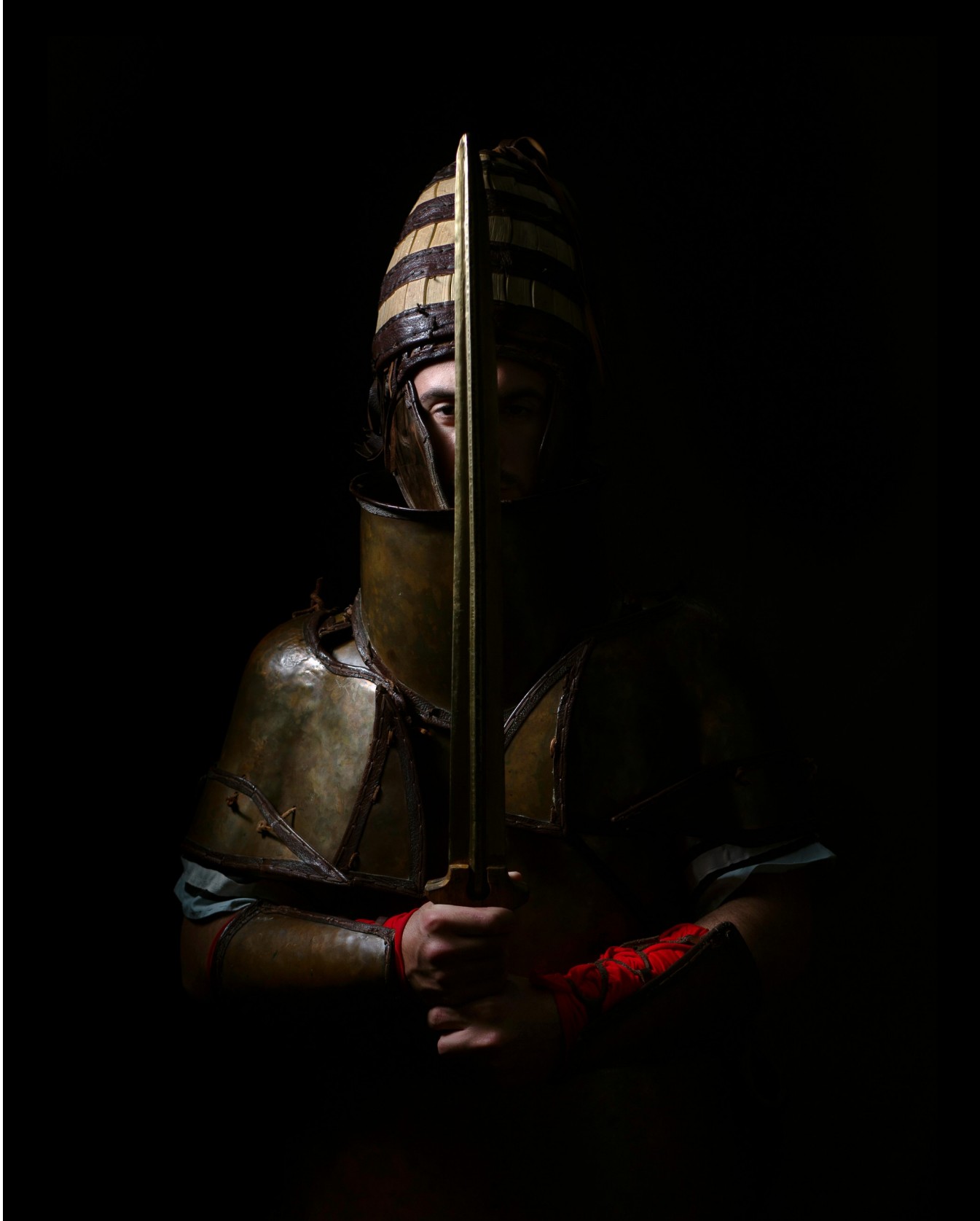

**Fig 7. Artistic photo taken after the end of the human experiment.** Photo credit: Andreas Flouris and Marija Marković. Permission required for reproduction.

of complex military practices. Therefore, we complemented our thematic analyses with an extensive review of the scholarly literature to ensure that our findings were in line with the best available evidence.

Our hypothetical date for the Trojan War (1300–1200 BC) was needed to enable estimates of the environmental conditions (temperature, day-night cycle, etc.) but should not be considered as a contribution to the continuing discussions about the date of the War. Moreover, the Trojan War cannot be taken as a historical event, at least in the form described in the Homeric epics [24]. There is no way to know whether it reflects a single event or a period of turmoil that was triggered during an alleged migration from Greece to the eastern Aegean, a narrative that has been also put into question [19,24,56].

Another important issue is the accuracy of the replicas of armour and weapons. Even slight variations in weight, balance, material, and craftsmanship can lead to different outcomes or misinterpretations of how these items were used in warfare. We carefully studied the Dendra armour (Online Supplement: Section 4.5) as well as the weapons used by Mycenaeans (Online Supplement: Section 4.6) and spent extensive efforts into building replicas that were as similar as possible to originals. However, it is important to acknowledge that some aspects of ancient warfare technology may be impossible to fully recreate or to understand them using modern replicas and experiments.

The physical condition of the participants in our human experiment is also an important factor to consider. We recruited Marines from the Hellenic Armed Forces who fit as much as possible the age and anthropometric characteristics of the elite warriors described in the Iliad (Online Supplement: Section 2.5). However, modern humans differ in physical condition, training, and endurance from ancient warriors. This difference may have affected the outcomes of our experiments, but this effect would further reinforce our findings as ancient warriors were likely more accustomed to the physical demands of their armour and weapons. Nevertheless, it is important to acknowledge that our Marines were a limited number, which may not represent the diversity of individuals involved in ancient warfare, and they lacked the combat experience and tactical knowledge of Late Bronze Age warriors. This gap can influence how weapons and armour were used in our study. Ensuring the safety of our participants was paramount, which means that full-scale, realistic combat scenarios could not be replicated accurately. While ethical considerations prevented us from replicating the true dangers and injuries of ancient warfare, we addressed a number of these issues by supplementing our experiments with the numerical simulation study, developing the Late Bronze Age Warrior model. Finally, although we gathered extensive evidence on the environmental conditions during the later phases of the Late Bronze Age and we further extended our investigations using numerical simulations, the specific environmental conditions (terrain, weather, etc.) of the ancient battlefield is challenging to replicate in a controlled laboratory setting.

All these limitations, however, with the exception of the improbable scenario 6, indicate that our conclusions about the utility of the Dendra armour err on the side of caution not optimism. If anything, it was even more practical in combat contexts than we have been able to demonstrate.

## Conclusions and significance

The discovery of the Dendra panoply in 1960 immediately raised the issue of its function. Could it have been worn in battle [13]–as repeatedly demonstrated by the exploits of the warriors in the Iliad? Was its use restricted to those who rode to battle in chariots? [57]–again, a regular practice in the epic and indicated in the Linear B archives by the association of armour and chariots. Or was it too cumbersome to be worn except on ceremonial occasions [2–4], as

stated, for example, in the caption to the pieces of armour from Thebes on display in the Archaeological Museum of Thebes? Only experiment–bound by strict criteria and content–with a replica could answer these questions since the metal of the panoply itself is too fragile for use. The series of archeo-physiological studies described here, merging knowledge in archaeology, history, human physiology, and numerical simulation, provide support for our conclusion that the Mycenaean armour found at Dendra was suitable for extended use in battle.

The Dendra armour changed our understanding of the ancient world in two important respects. First, it had long been known that early Greek warriors, the Mycenaeans, were well supplied with weapons but no armour had been found until that moment [58]. Second, it disproved the popular assertion that the references to bronze armour in Homer's Iliad were *later* interpolations so that his epic heroes would resemble the typical armour-wearing 'hoplite soldiers' of Archaic and Classical Greece [59].

The military aspects of Mycenaean society have long been evident from the massive fortifications which have never been lost from view [60] and from the burials of a warrior elite first revealed by Schliemann's excavations at Mycenae itself [61]. Swords, spears, and the plates from boar's tusk helmets are regular finds in tomb contexts and frequently illustrated in wall paintings and on seal stones. Shields and chariots are often illustrated but leave little trace in other aspects of the archaeological record. Body armour, as worn by Homer's warriors of the heroic age, was long regarded as poetic licence–or even later interpolation (Online Supplement: Sections 1.1–3).

The sketches of armour on the linear B tablets (S2 Fig in S1 File), first found at Knossos and later at Pylos, remained enigmatic. Were they really armour and, if so, was it the linen armour known to Classical scholars as *linothorax*? Ventris' decipherment of the script on the tablets and demonstration that it represented an early form of Greek hinted at the existence of bronze armour [62], but most scholars dismissed the possibility until the discovery of the Dendra armour. Even the identification of another dozen or so examples at different sites in Crete and mainland Greece (S1 Fig in S1 File) did not convince that it was either practical or common. The lists of numerous examples from the Room of the Chariot Tablets at Knossos were dismissed by the greatest authorities [63]. It seemed inconceivable that there could be so many purely ceremonial suits of bronze armour.

Sixty years on from the discovery of the Dendra armour, we now recognise that the records represent equipment provided to elite warriors by the palatial centres. Sixty years on we now understand, despite its cumbersome appearance at first sight, that it is not only flexible enough to permit almost every movement of a warrior on foot but also resilient enough to protect the wearer from most blows [4,5]. In addition, our experiments have now demonstrated, we hope convincingly, that the armour is also of a weight and structure to permit extended use in combat, day after day, for up to 11 hours, without detriment to a fit warrior (although, apart from the Iliad, we have no accounts of battles of such duration).

The Linear B tablets, with their records of repair and refurbishment as well as of issue to individual warriors, are surely the clearest evidence that the palatial centres maintained and equipped substantial fighting forces on stand-by, presumably, against time of need. The surviving Knossos record, which can only be partial because of the chance preservation of the tablets, indicates between 450 and 600 well-equipped individuals with both armour and chariots [64]. The resources committed to equipping such an 'army' (as well as those on the Greek mainland devoted to constructing vast fortifications) show that defence–against foes unknown to us–was among the most pressing concerns of the palatial authorities (whether or not we think of a Minos or an Agamemnon as leader) (Online Supplement: Sections 1.1–3).

Did they also have aggression in mind–the expansion of their territory? According to Hittite records, the Ahhiyawa–whom we recognise as Mycenaeans–had a sufficient presence in western Asia Minor in the second half of the 2nd Millennium BC to have both campaigned against the Hittites and later to become reconciled. At one point they were at war over 'Wilusa' in the North-West corner of Asia Minor–and there are strong linguistic arguments for equating Wilusa with Greek Ilion = Troy. At another moment, there are references by the Hittite king to the king of the Ahhiyawa as 'my brother'. It is clear from both the literary and the archaeological evidence that Millawanda = Miletus was for some time a Mycenaean stronghold [65,66].

Given that the Hittite kingdom dominated most of Anatolia and, at times, the northern parts of Syria and Mesopotamia [67], we must understand that only a significant military force could oppose them or gain such respect as recorded in the Hittite archives. We may reasonably ask whether the possession of full armour, as represented by the Dendra panoply, was perhaps the principal factor in gaining this respect. As far as we know, the Hittite warriors did not have an equivalent.

Another of the puzzles presented by the Linear B records is the chariots recorded on the same tablets as the armour (Online Supplement: Sections 1.2–3, S2 Fig in S1 File). Most experts regard the terrain of Greece, with a very few exceptions, as totally unsuitable for the use of battle-chariots. Although there are regular representations of chariots used in processions and a few in combat scenes, the numbers indicate a highly mobile force of armour-clad warriors. Where might these be deployed? Homer is familiar with the use of chariots to bring armoured warriors into battle, but they do not seem to be part of later hoplite warfare. May we assume this is a genuine memory of Bronze Age practice?

In the Near East chariot warfare had a long tradition [68]. At the battle of Kadesh in c 1274 BC, for example, both the Egyptians and the Hittites fielded 2000 or more chariots according to documentary evidence [69]. In later periods Greeks were much in demand as 'mercenaries'–in Egypt (Herodotus Histories 2.152) and the Persian empire (Xenophon Anabasis). Could this be an explanation for the presence of the image of a boar's tusk helmet on a scrap of 14th century BC Egyptian papyrus [70]? Might we perhaps think of the Knossos records as the response to a request for the services of a formidable 'battle group' by one of the Eastern Mediterranean powers—services for which the palace authorities would be well reimbursed?

Many questions remain unanswered, although today it is recognised that 'diplomatic' relations across the eastern Mediterranean included Mycenaean territory [71]. The evidence presented here that Mycenaean armour was suitable for extended use in battle enriches both our understanding of ancient Greek and Late Bronze Age warfare and the potential contexts in which it took place. It supplies a robust scaffold to support fresh research regarding the events that have underpinned social transformations in both prehistoric Europe and the Eastern Mediterranean. Our results sustain the notion that the Mycenaeans had a powerful impact in the Eastern Mediterranean (as confirmed in Hittite and Egyptian documents) [65,66,70] due, at least in part, to their armour technology. This may also help shed much-needed light on one of history's most frightful turning points: the collapse of the Eastern Mediterranean Bronze Age civilisations towards the end of the 2nd Millennium BC; a time of destruction and upheaval that marked the beginning of the Age of Iron.

## Supporting information

**S1 File. Details on the rationale, methods, and results of the study.**
(ZIP)

## Acknowledgments

We are indebted to Dimitrios Flouris and Michael Tzekakis for their advice and technical assistance during the thematic analyses of the Iliad. We wish to thank Dr Marija Marković and Mr Stelios Tsitimis for coordinating and executing the artistic photo shoot of the Dendra armour. We also thank Minos Koutedakis for his insightful suggestions during the design of the human experiment. We record with great sadness the recent death of Diana Wardle whose chance dinner-table conversation with Y.K. inspired our interdisciplinary project. *In pace requiesce.*

## Author Contributions

**Conceptualization:** Andreas D. Flouris, Tiago S. Mayor, Yiannis Koutedakis, Ken Wardle, Diana Wardle.

**Data curation:** Andreas D. Flouris, Stavros B. Petmezas, Panagiotis I. Asimoglou, João P. Vale, Tiago S. Mayor.

**Formal analysis:** Andreas D. Flouris, João P. Vale, Tiago S. Mayor.

**Investigation:** Andreas D. Flouris, Stavros B. Petmezas, Panagiotis I. Asimoglou, Tiago S. Mayor, Ken Wardle.

**Methodology:** Andreas D. Flouris, Stavros B. Petmezas, Panagiotis I. Asimoglou, João P. Vale, Tiago S. Mayor, Giannis Giakas, Athanasios Z. Jamurtas, Yiannis Koutedakis, Ken Wardle, Diana Wardle.

**Project administration:** Andreas D. Flouris, Stavros B. Petmezas, Athanasios Z. Jamurtas.

**Resources:** Andreas D. Flouris.

**Software:** João P. Vale, Tiago S. Mayor.

**Supervision:** Andreas D. Flouris.

**Validation:** Andreas D. Flouris, João P. Vale, Giannis Giakas, Ken Wardle, Diana Wardle.

**Visualization:** Andreas D. Flouris, João P. Vale, Diana Wardle.

**Writing – original draft:** Andreas D. Flouris, Ken Wardle.

**Writing – review & editing:** Andreas D. Flouris, Stavros B. Petmezas, Panagiotis I. Asimoglou, João P. Vale, Tiago S. Mayor, Giannis Giakas, Athanasios Z. Jamurtas, Yiannis Koutedakis, Ken Wardle, Diana Wardle.

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
