## [Decision Letter · Decision Letter 0]

5 Sep 2023

PONE-D-23-17949Analysis of prehistoric combat in full body armour based on physiological principles: a series of studies using thematic analyses, a cross sectional human trial, and numerical simulationsPLOS ONE

Dear Dr. Flouris,

Thank you for submitting your manuscript to PLOS ONE. After careful consideration, we feel that it has merit but does not fully meet PLOS ONE’s publication criteria as it currently stands. Therefore, we invite you to submit a revised version of the manuscript that addresses the points raised during the review process.

The attached reviews provide several important observations and questions. In light of these reviews, the manuscript needs some revision. Please consider the comments of both Reviewer 1 and 2 while revising your manuscript.

Furthermore, apart from deleting the last sentence of your paper ("…and set the origins of Western Civilization…"), as noted by one of the reviewers, I further recommend to delete the sentence “Eratosthenes provided…13^th^ century BC” (lines 220–224). Later narratives cannot be taken as reliable evidence for the dating of the Trojan War. The Trojan War cannot be taken as a historical event, at least in the form described in the Homeric epics. There is quite a long list of literature on the historicity of the Trojan War and the historical implications of early Greek poetry. There is also no way to know if it reflects a single event or a turmoil period that was triggered during an alleged migration from Greece to the eastern Aegean, a narrative that has been also put into question. Additionally, I recommend to avoid dating the event in the 13^th^ (throughout the paper) or any other century and replace that with relative dates (e.g. in the later phases of the Late Bronze Age, etc.). The above mentioned flaws undermine the paper‘s value.

It is not clear how information provided by Strabo can contribute to the reconstruction of the Trojan landscape during the Late Bronze Age period (lines 214, 238). Please explain or rephrase the sentence.

It is advisable to avoid “accurate” in any description of replicas (lines 299-300).

Please explain why you use methods of the Archaic period to reconstruct a Late Bronze Age sword (line 182). Additionally, there were no cooper but bronze weapons as one of the reviewers already noticed. This does not certainly affect the – indeed innovative – results of your study but you have to comment on that.

Please remove the photo from page 8. You can either include it in the figures cited in the text or delete it. My suggestion would be to include more photos from the experimental study in the text (as you do in the supporting information document).

I finally suggest that you include more recent literature on early Aegean warfare (see, e.g.,  Heckel et al. A Companion to Greek Warfare, 2021, with further literature).

We look forward to receiving your revised manuscript.

Kind regards,

Stefanos Gimatzidis, Ph.D.

Academic Editor

PLOS ONE

Journal Requirements:

2. We note that Figures 1 and S9 in your submission contain copyrighted images. All PLOS content is published under the Creative Commons Attribution License (CC BY 4.0), which means that the manuscript, images, and Supporting Information files will be freely available online, and any third party is permitted to access, download, copy, distribute, and use these materials in any way, even commercially, with proper attribution. For more information, see our copyright guidelines: http://journals.plos.org/plosone/s/licenses-and-copyright.

a. You may seek permission from the original copyright holder of Figures 1 and S9 to publish the content specifically under the CC BY 4.0 license. 

3. We note that Figures 2 and S7 in your submission contain map/satellite images which may be copyrighted. All PLOS content is published under the Creative Commons Attribution License (CC BY 4.0), which means that the manuscript, images, and Supporting Information files will be freely available online, and any third party is permitted to access, download, copy, distribute, and use these materials in any way, even commercially, with proper attribution. For these reasons, we cannot publish previously copyrighted maps or satellite images created using proprietary data, such as Google software (Google Maps, Street View, and Earth). For more information, see our copyright guidelines: http://journals.plos.org/plosone/s/licenses-and-copyright.

a. You may seek permission from the original copyright holder of Figures 2 and S7to publish the content specifically under the CC BY 4.0 license.  

5. We note that Figures 3, S4 and S 17 includes an image of a participant in the study. 

Reviewers' comments:

Reviewer's Responses to Questions

**Comments to the Author**

1. Is the manuscript technically sound, and do the data support the conclusions?

Reviewer #1: Yes

Reviewer #2: Yes

2. Has the statistical analysis been performed appropriately and rigorously? 

Reviewer #1: Yes

Reviewer #2: Yes

3. Have the authors made all data underlying the findings in their manuscript fully available?

Reviewer #1: Yes

Reviewer #2: Yes

4. Is the manuscript presented in an intelligible fashion and written in standard English?

Reviewer #1: Yes

Reviewer #2: Yes

5. Review Comments to the Author

Reviewer #1: This is the third time I have written this on this form. The previous attempts seems not to have recorded by your system.

This is an interesting and important study. Previous interpretations of the Dendra Corselet have been speculative and devoid of any real analysis. Therefore the literature about the Dendra armour has tended to be full of "truisms". The main "truism" is that the Dendra armour is essentially ceremonial and could not have been used practically. The author of this article sets out to demonstrate this is not so. Using a series of practical scientific experiments he establishes that the Dendra armour is practical and could have worn in actual combat. This is the primary aim of the project, and is discussed at the beginning of the article. The methodology for the project is described as a series of practical experiments wherein an accurate replica of the armour is worn by a group of volunteers (fit young men from the Greek army), and they undertook physical actions appropriate to combat. They were then subjected to a series of the tests to establish their physiological response to the stresses of the actions. In the article the tests are described fully and the results presented (additional detail is also provided through links to online databases). Consequently the primary of aim of the project is proved: that the Dendra armour could be used practically in combat and did not impeded the warrior's mobility or combat effectiveness.

That said: the article as presented is essentially quantitative in its discussion. This is one of its strengths. It is written seemingly for a scientifically minded audience, with appropriate "hard" evidence. The author however, in part of the online material includes a supplementary discussion of the project which is primarily qualitative in form and style. It is therefore more accessible to those of us with less scientific minds. It manages to address the sorts of questions which might occur to archaeologists and scholars of ancient combat. It also elaborates more on the interpretative implications of the study. Therefore I strongly urge the author to consider incorporating some of the discussion from the qualitative version into the quantitative. Or he could design the qualitative version to stand as a Part 2. It will definitely have a wider accessibility. I am happy to recommend publication of this article.

Reviewer #2: I congratulate the authors on a most original, thorough, and thought-provoking piece of research, which adds significantly to our knowledge and understanding of Mycenaean combat tactics. The research conclusively demonstrates that the Dendra armour was a functional, and probably much used, piece of defensive kit. What's more, the article and supplementary materials are written in a fluent and sophisticated language that does not require any further editing. I warmly recommend publishing the manuscript once the authors have carried out the minor corrections recommended below:

- line 78ff. The authors give the impression that the Iliad provides a coherent account of fighting tactics from the late Mycenaean period. As I'm sure they are aware, the Iliad is palimpsest of contents created in different periods, as shown in particular by mismatching or otherwise problematic descriptions of weapons, armour and tactics (e.g., the well-known example of chariots being used in the Iliad as 'battlefield taxis'). They should discuss, in at least a paragraph, how they navigated the problem and resolved the many contradictions posed by the text.

- line 181. It's not clear why the replica sword was manufactured from pure copper considering that late Mycenaean swords were made of tin-bronze. This should be corrected or explained.

- line 383: I strongly recommend that the problematic closing "and set the origins of Western Civilization" is deleted. The article closes rather well, and in fact better, with "the beginning of the Age of Iron".

6. PLOS authors have the option to publish the peer review history of their article (what does this mean?). If published, this will include your full peer review and any attached files.

Reviewer #1: **Yes: **Alan A.D. Peatfield

Reviewer #2: **Yes: **Andrea Dolfini

---

## [Author Response · Author response to Decision Letter 0]

3 Mar 2024

Please, see the submitted document.

---

## [Editor Report · Decision Letter 1]

18 Mar 2024

Analysis of Greek prehistoric combat in full body armour based on physiological principles: a series of studies using thematic analysis, human experiments, and numerical simulations

PONE-D-23-17949R1

Dear Dr. Flouris,

We’re pleased to inform you that your manuscript has been judged scientifically suitable for publication and will be formally accepted for publication once it meets all outstanding technical requirements.

Kind regards,

Stefanos Gimatzidis, Ph.D.

Academic Editor

PLOS ONE
---

## [Editor Report · Acceptance letter]

30 Apr 2024

PONE-D-23-17949R1 

PLOS ONE

Dear Dr. Flouris, 

I'm pleased to inform you that your manuscript has been deemed suitable for publication in PLOS ONE. Congratulations! Your manuscript is now being handed over to our production team.

Kind regards, 

on behalf of

Dr. Stefanos Gimatzidis 

Academic Editor

PLOS ONE